# The Management of Diabetes Mellitus Using Medicinal Plants and Vitamins

**DOI:** 10.3390/ijms24109085

**Published:** 2023-05-22

**Authors:** Clement G. Yedjou, Jameka Grigsby, Ariane Mbemi, Daryllynn Nelson, Bryan Mildort, Lekan Latinwo, Paul B. Tchounwou

**Affiliations:** 1Department of Biological Sciences, College of Science and Technology, Florida Agricultural and Mechanical University, 1610 S. Martin Luther King Blvd, Tallahassee, FL 32307, USA; lekan.latinwo@famu.edu; 2Department of Biological Sciences, School of Arts and Sciences, Alcorn State University, 1000 ASU Drive, Lorman, MS 39096, USA; 3Department of Biology, College of Science, Engineering and Technology, Jackson State University, 1400 Lynch Street, Box 18750, Jackson, MS 39217, USA; 4Department of Health Administration, Morehouse School of Medicine, 720 Westview Dr. SW, Atlanta, GA 30310, USA; 5Department of Pharmaceutical Sciences, College of Pharmacy, Howard University, 2400 6th St, NW, Washington, DC 20059, USA; bryan.mildort@bison.howard.edu; 6RCMI Center for Urban Health Disparities Research and Innovation, Morgan State University, 1700 E. Cold Spring Lane, Baltimore, MD 21252, USA

**Keywords:** diabetes mellitus, medicinal plants, vitamins, phytochemicals, anti-diabetic, treatment

## Abstract

Diabetes mellitus (DM) is a serious chronic metabolic disease that is associated with hyperglycemia and several complications including cardiovascular disease and chronic kidney disease. DM is caused by high levels of blood sugar in the body associated with the disruption of insulin metabolism and homeostasis. Over time, DM can induce life-threatening health problems such as blindness, heart disease, kidney damage, and stroke. Although the cure of DM has improved over the past decades, its morbidity and mortality rates remain high. Hence, new therapeutic strategies are needed to overcome the burden of this disease. One such prevention and treatment strategy that is easily accessible to diabetic patients at low cost is the use of medicinal plants, vitamins, and essential elements. The research objective of this review article is to study DM and explore its treatment modalities based on medicinal plants and vitamins. To achieve our objective, we searched scientific databases of ongoing trials in PubMed Central, Medline databases, and Google Scholar websites. We also searched databases on World Health Organization International Clinical Trials Registry Platform to collect relevant papers. Results of numerous scientific investigations revealed that phytochemicals present in medicinal plants (*Allium sativum*, *Momordica charantia*, *Hibiscus sabdariffa* L., and *Zingiber officinale*) possess anti-hypoglycemic activities and show promise for the prevention and/or control of DM. Results also revealed that intake of vitamins C, D, E, or their combination improves the health of diabetes patients by reducing blood glucose, inflammation, lipid peroxidation, and blood pressure levels. However, very limited studies have addressed the health benefits of medicinal plants and vitamins as chemo-therapeutic/preventive agents for the management of DM. This review paper aims at addressing this knowledge gap by studying DM and highlighting the biomedical significance of the most potent medicinal plants and vitamins with hypoglycemic properties that show a great potential to prevent and/or treat DM.

## 1. Introduction

Diabetes mellitus (DM) is a metabolic disorder that leads to chronic hyperglycemia, a pathogenesis condition that may include defects in insulin secretion and/or action [1,2]. It is estimated that one in three Americans will develop diabetes sometime in their lifetime [3]. The most common form of DM is type 2 diabetes mellitus (T2DM), which accounts for approximately 90% of DM cases. T2DM is predominantly due to the failure of the bodily tissues to respond to insulin or synthesize enough insulin [4,5]. Several scientific studies have indicated that diabetes affects the human quality of life by causing major risk factors for adverse complications such as stroke, amputation, kidney failure, and blindness, leading to significant morbidity and premature mortality [6,7,8]. 

As seen in Figure 1, the International Diabetes Federation (IDF) estimated that there were approximately 463 million adults with diabetes in 2019, which have been projected to raise up to 578 million adults by 2030 and 700 million by 2045 [9]. 

The treatment strategies for DM have improved over the last few decades. However, anti-diabetic drugs have serious effects such as hypoglycemic coma and liver and kidney disorders [10]. The World Health Organization (WHO) recommends the use of medicinal plants in food items for the treatment of DM [11,12]. At least four billion people living in developing countries use medicinal plants for the treatment of metabolic diseases such as DM [13,14]. Therefore, medicinal plants, vitamins, and essential elements with anti-hypoglycemic properties remain essential for the management of diabetes. Scientific reports showed that medicinal plants, vitamins, and essential elements have been successfully used to lower the blood sugar level in the shape of pre-clinical and clinical studies [15,16]. For example, A study showed that zinc intake regulates insulin receptors and extents insulin action [17]. A study showed that garlic provides a protective effect against diabetic retinopathy in adult albino rats [18]. A number of phytochemicals that have anti-diabetic properties present in medicinal plants have been discovered based on differences in chemical structure and have been classified as major groups [19,20]. The major groups of phytochemicals are alkaloids, aromatic acids, carotenoids, coumarins, essential oils, flavonoids, glycosides, organic acid, phenols and phenolics, phytosterols, protease inhibitors, saponins, steroids, tannins, terpenes, and terpenoids [21,22,23]. Recent pharmacological studies have revealed the anti-diabetic properties of medicinal plants and vitamins including anti-hyperglycemic, anti-lipidemic, hypoglycemic, and insulin mimicking [24,25].

The research objective of this review article is to study DM and explore the available treatments for this disease based on medicinal plants and vitamins.

## 2. Approaches

### Data Sources and Data Extraction

Systematic literature searches were conducted on peer-reviewed publications using key terms including the medicinal plants, *Allium sativum* (Garlic); *Momordica charantia* (Bitter Melon); *Hibiscus sabdariffa* L. (Roselle Plant); *Zingiber officinale* Rosc (Ginger); Vitamins C, D, and E; diabetes mellitus; and the prevention and treatment of diabetes mellitus. These literature searches were performed in five databases including Science Direct, PubMed Central, MEDLINE, Google Scholar, and trusted governmental agencies (World Health Organization, Food and Drug Administration, and Center for Disease Control and Prevention). Only peer-reviewed articles in which the titles and abstracts involved the previously listed key terms were selected. Publication selection included articles available from 2010 to 2022.

## 3. Results and Discussions

We found several peer-reviewed articles that addressed the health benefits of medicinal plants and vitamins for the management of diabetes. The summary results and discussions of this paper give an update of the overview of medicinal plants including *Allium sativum* (Garlic); *Momordica charantia* (Bitter Melon); *Hibiscus sabdariffa* L. (Roselle Plant); *Zingiber officinale* Rosc (Ginger); Vitamins C, D, and E; and new therapeutic approaches that aim at preventing and treating DM. In our results and discussion sections, we did not summarize all the therapeutic approaches investigated in each study. However, we highlighted their main findings in the present article. These are discussed in more detail in subsequent sections below.

### 3.1. Medicinal Plants and Their Anti-Diabetic Properties

Medicinal plants or plant-based medicine has been used cost-effectively throughout the world to prevent and/or treat diabetes. In fact, many developing countries rely on plant-based medicine to treat people with diabetes and other conditions. Several pharmaceuticals commonly used today are structurally derived from natural compounds that are found in traditional medicinal plants. For example, the anti-hyperglycemic drug called metformin, currently used to treat diabetes, can be traced back to the traditional use of *Galega officinalis* to treat diabetes [26,27]. Most commonly used medicinal plants and vitamins with hypoglycemic activities to improve the immune system and manage blood sugar levels in humans include *Allium sativum* (garlic), *Momordica charantia* (Bitter Melon), *Hibiscus sabdariffa* L. (Roselle Plant), *Zingiber officinale* Rosc (Ginger), and Vitamins C, D, and E (Figure 2, and Table 1). Given that many medicinal plants are easily accessible, cheap, and useful for the management of diabetes, many developing countries and a few wealthy countries use medicinal plants to meet their healthcare needs.

#### 3.1.1. *Allium sativum* and Its Application to Diabetes Mellitus

*Allium sativum* is the scientific name for garlic, which is a plant species in the family of Amaryllidaceae. Garlic is known to have various health benefits such as lowering cholesterol, improving blood pressure, and boosting the immune system. Garlic has also been shown to have potential benefits for individuals with diabetes. Research has suggested that garlic may help lower blood sugar levels and improve insulin sensitivity, which can be beneficial for diabetes.

The major phytochemicals present in garlic include (1) allicin, which is one of the most well-known phytochemicals in garlic and is responsible for its pungent odor; (2) sulfur compounds, including diallyl disulfide and diallyl trisulfide, which have antioxidant, anti-inflammatory, anti-diabetes, and anti-cancer properties [28,29]; (3) flavonoids, including quercetin and kaempferol, which are known for their antioxidant properties; (4) saponins, which are natural detergents that have cholesterol-lowering properties; and (5) fructans, which are a type of carbohydrates that can act as prebiotic candidates. 

The beneficial health effects of garlic include its anti-inflammatory, immunomodulatory, cardioprotective, hypolipidemic, hypoglycemic, antioxidant, antibiotic, antifungal, antimicrobial, antiseptic, anticancer, and antiviral activities [28,30,31]. It has been demonstrated clinically that garlic supplementation with standard anti-diabetic drugs provides diabetic control in type 2 diabetes [32]. In addition, clinical trials have demonstrated that garlic and garlic derivatives reduce insulin resistance effectively [33,34]. Furthermore, garlic component acts as hydrogen sulfur donors that control type 2 diabetes [29]. Another study has demonstrated garlic reduces lipid profile and glucose parameters such as fasting glucose levels and hemoglobin A1c (HbA1C) in diabetic patients [35].

#### 3.1.2. *Momordica charantia* and Its Application to Diabetes Mellitus

*Momordica charantia*, known as bitter melon is used as a complementary or alternative therapy for the treatment of DM in both developing and wealthy countries. It contains compounds that are effective in regulating and lowering blood glucose levels in patients with DM. It regulates and lowers blood glucose in diabetes patients by improving insulin sensitivity and reducing glucose production in the liver [36].

The beneficial health effects of bitter melon include its anti-inflammatory, immunomodulatory, hypolipidemic, hypoglycemic, antioxidant, antifungal, antibacterial, anticancer, and antiviral activities [37,38,39]. The phytochemical analysis of the leaf, fruit, and seed of bitter melon shows the presence of amino acids, carbohydrates, flavonoids, glycosides, minerals, phenols, phytosterols, saponins, tannins, and vitamins which are responsible for anti-oxidants, anti-inflammatory, immunomodulatory, hypolipidemic, and anti-hyperglycemic activities [39,40,41].

Studies showed that hypoglycemic herbs increase insulin secretion, enhance glucose intake by adipose or muscle tissues, and inhibit glucose absorption from the intestine and glucose production from the liver [42,43]. Several in vivo studies using animals indicated that bitter melon has hypoglycemic effects which stimulate glucose uptake into skeletal muscle cells and increase insulin secretion [38,44,45]. Similarly, a few clinical reports showed that bitter melon effectively lowers glucose levels in patients with type 2 diabetes [46,47,48]. For example, Kim and collaborators performed a randomized, placebo-controlled study. Blood glucose levels, lipid profiles, and adverse events were investigated after 12 weeks of treatment. Ninety subjects were included in the final analysis for the glucose-lowering efficacy of bitter melon. Results showed that there were no differences in age, sex, or glycated hemoglobin (HbA1C) levels between the bitter melon extract and placebo groups. After treatment with bitter melon extract for 12 weeks, the HbA1c levels of the bitter melon and placebo groups remained unchanged; however, the average fasting glucose level of the bitter melon group decreased (*p* = 0.014). No serious adverse events were reported during the treatment period. Results proved that bitter melon has effects of glucose-lowering in patients with type 2 diabetes [48]. Another study showed that bitter melon permanently normalized blood glucose levels in diabetic rats compared to healthy rats [41].

#### 3.1.3. *Hibiscus sabdariffa* L. (Roselle) and Its Application to Diabetes Mellitus

*Hibiscus sabdariffa* L. (roselle) is a plant belonging to the Malvaceae family, growing wild in tropical climates in many countries [49]. It has been used in traditional medicine for many years due to its high content of pharmacologically active compounds and good healing properties [50,51]. The most reported health-beneficial effects of *Hibiscus sabdariffa* L. (roselle) include its anti-hypertensive, anti-inflammatory, body fat mass reduction, immunomodulatory, hypoglycemic, antioxidant, lipid-lowering, anticancer, and anti-xerostomic effects [52,53,54]. These therapeutic effects of *Hibiscus sabdariffa* L. (roselle) have been associated with the presence of bioactive and functional compounds such as phenolic acids, flavonoids, anthocyanins, organic acids, and dietary fiber [55,56]. An animal study demonstrated that the oral administration of *Hibiscus sabdariffa* L. (roselle) at doses of 72 mg/200 g body weight and 288 mg/200 g body weight for 21 days lowered blood glucose levels in rats with chronic diabetes [57]. Harrison and colleagues evaluated the effect of *Hibiscus sabdariffa* tea (10 g of powder in 500 mL of boiling water) intake in controlling post-prandial blood glucose (60 min) levels of one volunteer for six consecutive days. They found that *Hibiscus sabdariffa* tea reduces the rise in blood glucose and reduces post-prandial hyperglycemia [58].

#### 3.1.4. *Zingiber officinale* and Its Application to Diabetes Mellitus

*Zingiber officinale*, known as ginger, is a widely used flavor for various foods and drinks. It has been used as an herbal remedy to treat various ailments worldwide since ancient times. Phytochemicals analysis revealed that *Zingiber officinale* possesses phenolic compounds such as gingerols, shogaols, paradols, and non-volatile compounds, including zingiberone, zingiberole, and zingiberene [59,60]. 

The beneficial health effects of ginger include its anti-inflammatory, immunomodulatory, antioxidant, hypolipidemic, hypoglycemic, and antiemetic effects as well as lowering blood pressure and blood sugar [61,62,63]. Many studies in both animals and humans have demonstrated that ginger is a hypoglycemic food adjunct with promise for the treatment of type 2 diabetes in human subjects [64,65,66]. Ginger exerts its mechanism of action by modulating insulin release, promoting glucose clearances in insulin-responsive peripheral tissues, which is crucial in maintaining blood glucose homeostasis [67]. Furthermore, it has been reported that 6-gingerol increases the glucose uptake at insulin-responsive adipocytes and shows that insulin-responsive glucose uptake has increased and improved diabetes in cells treated with gingerol [60]. A previous study indicated that the administration of 50% ethanolic extract of ginger rhizomes prevents the development of obesity and insulin resistance in rats inter alia by regulating the PPAR receptors [68].

### 3.2. Vitamins and Their Anti-Diabetic Properties

Vitamins exert important effects on the risk of DM as well as its progression and complications. The intake of Vitamins C, D, E, or a combination of them all has been associated with decreased risk of diabetes in the general population. For example, Vitamins C, D, or E has been hypothesized to exhibit anti-diabetic properties by regulating insulin secretion or insulin sensitivity, producing anti-inflammatory, immunomodulatory, antioxidant, hypolipidemic, and hypoglycemic effects [69,70,71].

#### 3.2.1. Vitamin D and Its Application to Diabetes Mellitus

Vitamin D (calciferol) is a fat-soluble vitamin that plays a role in the enhancement of the immune system, regulation of bone growth, and absorption of calcium, iron, magnesium, phosphate, and zinc [72,73]. Vitamin D (Vit D) exists in two forms including cholecalciferol (Vitamin D3) and ergocalciferol (Vitamin D2) [74,75]. It is found naturally in fish (salmon, tuna, sardines), dairy (milk), green (spinach, okra, kale), beans (soy and white), meat (beef liver), and exposure to ultraviolet B [74]. Skin exposure to solar ultraviolet B radiation synthesized Vitamin D3; meanwhile, Vitamin D2 is synthesized by plants [76,77]. The receptors for Vitamin D are found in most tissue or organs and are involved in several biological functions such as promoting calcium absorption in the gut, maintaining adequate serum calcium and phosphate concentration, reducing inflammation, and modulating several processes, including cell growth, immune function, glucose metabolism, and insulin sensitivity [78]. However, the impairment of pancreatic beta cells and insulin-resistance have been associated with a deficiency in Vitamin D [79,80].

Recent investigations have shown that low level of Vitamin D is associated with impaired fasting glucose, hypertension, obesity, glucose intolerance, and the development of T2DM [80]. Preclinical studies have demonstrated that pancreatic beta cell function properly with an adequate level of Vitamin D because it helps in promoting the conversion of proinsulin to insulin, increasing insulin output, and enhancing insulin action through the regulation of the calcium pool [81,82,83]. Vitamin D further serves as a chemical messenger and is involved in the regulation of transcription such as the down-regulation of pro-inflammatory cytokine genes such as Interleukin-2, interleukine-12, tumor necrosis factors -α, production of anti-inflammatory cytokines, and protection of beta-cell destruction [82]. A randomized control double-blind intervention study noted a significant improvement of insulin sensitivity in diabetic patients supplementing 4000 IU of Vitamin D for 6 months compared to a placebo [84]. A similar study also noted that Vitamin D supplements affect insulin secretion in prediabetics patients compared to control [85]. Furthermore, Vitamin D supplements are associated with a reduction in the level of metabolic parameters, including total cholesterol, low-density lipoprotein, glycated hemoglobin, triglyceride, and diabetic complication [86,87].

#### 3.2.2. Vitamin E and Its Application to Diabetes Mellitus

Vitamin E is found mainly in plant-based oils (peanuts, olive, soybean oil), nuts (almonds), seeds (sunflower seeds), fruits (mango, red bell pepper), and vegetables (collard green, spinach, and beets green). It is a collective group of fat-soluble compounds with eight isoforms that can be categorized into tocopherol and tocotrienol isoforms. The tocopherol isoforms can be classified into alpha (α), beta (β), gamma (γ), and delta (δ) categories and have a saturated side and chain on the chromanol ring. Meanwhile, tocotrienol isoforms have an unsaturated side chain, and the two types can be further categorized into α, β, γ, and δ, and the α tocopherol best meets the dietary requirements of humans. Vitamin E is considered a powerful antioxidant that limits the production of ROS formed when fat undergoes oxidation and, therefore, helps prevent or slow chronic conditions associated with free radicals. Research has demonstrated that a high dose of vitamin E reduces oxidative stress biomarkers and increases immune defense. A randomized study in patients with diabetic nephropathy showed that supplementing 800 IU vitamin E for 12 weeks significantly increased the levels of glutathione peroxidase (GPx) compared to the placebo [88]. A similar prospective study on type 2 diabetics with or without complications supplementing 4000 IU of vitamin E along with hypoglycemic drugs daily for 9 months showed a gradual decrease in fasting blood sugar, serum glycated hemoglobin (HbA1C), and BMI compared to control [89]. In sum, the antioxidant properties of vitamin E have the potential to delay diabetic complications.

#### 3.2.3. Vitamin C and Its application to Diabetes Mellitus

Vitamin C, or ascorbic acid, is an antioxidant and plays several functions such as enzyme cofactors, radical scavengers, electron transport donors, or receptors in the plasma membrane [90]. Deficiency of Vitamin C leads to defective formation of collagen, blood vessels, and connective tissue in the bone, dentine, cartilage, skin, and oxidative stress [90]. Oxidative stress often leads to glucose metabolism and hyperglycemia. Hyperglycemia promotes the oxidation of glucose to form free radicals. The free radical generation above the scavenging potential of endogenous antioxidants may result in macro- and microvascular dysfunction [91]. Vitamin C biomolecules can protect from oxidation by participating in oxidation-reduction reactions, in which dehydroascorbic acid will be oxidized and reduced back into ascorbate [92]. The main sources of Vitamin C are fresh fruits, vegetables, and aromatic herbs [90]. The vernacular names of fruits with high contents of Vitamin C include the Kakadu plum from Australia, camu-camu from South America, fruit star, guava, kiwi, strawberry, orange, lemon, and pear [93,94]. The cruciferous vegetables and aromatic herbs expressing elevated levels of Vitamin C include broccoli, kale, pepper, cabbage, parsley, chives, and coriander [95]. Temperature plays an important role in Vitamin C preservation and stability. The gentle way to preserve Vitamin C content, and avoid possible leaching out into water, degradation, and pH changes, is steaming or boiling in a small quantity of water for very short-time and deep freezing for long-term storage [96,97]. As many fruits and vegetables contain Vitamin C, a prospective cohort study of 23,953 men who were diabetic-free as a baseline discovered that 1741 men who developed type 2 diabetes increased their vegetable and fruit intake to 1.6 servings per week (10) [98].

Mason et al. (2018) in a study found that type 2 diabetic patients supplementing ascorbic acid experienced a reduction in blood sugar as well as blood pressure in 4 months compared to placebo [99]. In addition, a cross-sectional study investigating the correlation between Vitamin C serum level and fasting blood sugar, glycated hemoglobin, serum malondialdehyde, and lipid levels in diabetic patients noted that low levels of Vitamin C significantly increase the systolic blood pressure, glycated hemoglobin, and malondialdehyde levels, leading to an increase in oxidative stress biomarkers [70]. The report also noted an inverse relationship between fasting blood sugar, total cholesterol, and Vitamin C levels [70]. A similar result was found in a retrospective study exploring Vitamin C levels, renal dysfunction, and obesity in patients with type 1 diabetes and type 2 diabetes [100]. To sum up, these findings suggest Vitamin C therapy to ameliorate glycemic and blood pressure in diabetic patients.

### 3.3. Medicinal Properties of Selected Medicinal Plants and Vitamins

The literature review revealed that *Allium sativum*, *Momordica charantia*, *Hibiscus sabdariffa* L., *Zingiber officinale*, and Vitamins (C, D, and E) have in common night (9) medicinal properties, including anti-diabetic, hypolipidemic, hypoglycemic, immunomodulatory, antioxidant, anti-inflammatory, anti-cancer, anti-bacterial, and anti-fungal properties (Figure 2 and Table 1).

**Figure 2 ijms-24-09085-f002:**
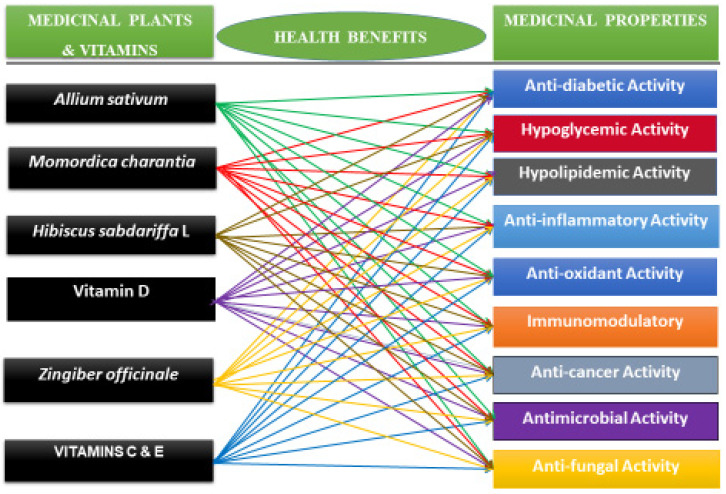
Summary of medicinal properties of *Allium sativum*, *Momordica charantia*, *Hibiscus sabdariffa* L., *Zingiber officinale*, and Vitamins (C, D, and E). The medicinal plants listed in Figure 2 are common herbs consumed worldwide as a functional food and traditional home remedies for the prevention and/or treatment of diabetes.

**Table 1 ijms-24-09085-t001:** Molecular mechanisms of action and clinical studies of medicinal plants (*Allium sativum*, *Momordica charantia*, *Hibiscus sabdariffa* L., *Zingiber officinale*, and Vitamins (C, D, and E) in the management of DM.

Medicinal Plants & Vitamins	Mechanisms of Action	Clinical Studies

*Allium Sativum* 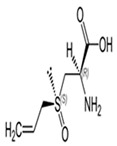	*Allium Sativum* plays a role in the treatment of diabetes by enhancing the gene expression of caspase 3 and caspase 9, reducing IL-1β, IL-6, and TNF-α level and increasing IFN-γ in vitro and in vivo [101].	A double-blind clinical trial in diabetic patients demonstrated garlic intake at a dose of 750 mg three times per day for 12 weeks had potential effects for treating diabetes by reducing fasting glucose blood levels through the decrease in hemoglobin A1c (HbA1C) [102].

Vitamin C 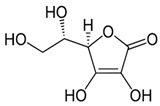	Vitamin C enhances the immune system by stimulating IFN production and lymphocyte proliferation, enhancing neutrophil phagocytic capability [103]. Vitamin C intake regulates fasting blood glucose (FBG) and glycosylated hemoglobin A1c (HbA1C) and improves insulin resistance [104,105].	More clinical trials are needed to confirm whether Vitamin C shows promise as an effective therapeutic agent for diabetes mellitus.

*Momordica charantia* 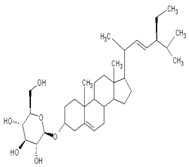	*Momordica charantia* exerts its hypoglycemic effect through multiple mechanisms of action including the stimulation or inhibition of the key enzymes of hexose monophosphate pathways. It can stimulate key enzymes of the hexose monophosphate pathway, inhibit glucose uptake by the intestine, increase the utilization of peripheral and skeletal muscle glucose, inhibit gluconeogenesis and adipocytes differentiation, and normalize the islet βcells [39,106,107,108].	Clinical trials are needed (clinical studies of *Momordica charantia* for the treatment of diabetes have been sparse and sporadic).

*Hibiscus sabdariffa* 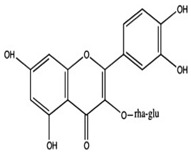	The mechanism of action of *Hibiscus sabdariffa* is based on the strong ability to delay the digestion of complex sugars into simple sugars, reduce the absorption of simple sugar, and lower total blood glucose [109,110].	Clinical trials are needed.

Vitamin D 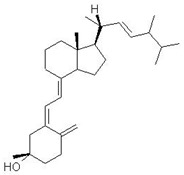	Studies showed that Vitamin D promotes the conversion of proinsulin to insulin, increases insulin output, and enhances insulin action through the regulation of the calcium pool [81,82].	A randomized control double-blind intervention study noted a significant improvement of insulin sensitivity in diabetic patients supplementing 4000 IU of Vitamin D for 6 months compared to placebo [84].

Vitamin E 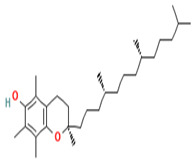	Animal models and human studies have demonstrated that vitamin E intake blocks LDL lipid peroxidation, prevents the oxidative stress linked to T2DM-associated abnormal metabolic patterns (hyperglycemia, dyslipidemia, and elevated levels of FFAs), and, subsequently, attenuates cytokine gene expression.	A recent report evaluated the effects of a combination of Vitamin C (1000 mg/day) and vitamin E (400 IU/day) for four weeks on insulin sensitivity in untrained and trained healthy young men and concluded that such supplement may preclude the exercise-induced amelioration of insulin resistance in humans [111].

*Zingiber Officinale* 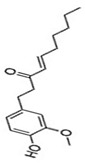	Diabetic patients who took 2 g of ginger for 2 months revealed a reduction in insulin, homeostasis model assessment (HOMA), and low-density lipoprotein (LDL), with no impact on fasting plasma glucose (FPG), HbA1C, total cholesterol, and HDL levels [112].	A double-blinded placebo-controlled randomized clinical trial conducted on two groups of patients with type 2 diabetes reported that ginger powder improved glycemic indices as well as TAC and PON-1 activity in patients [64]. However, more clinical trials are needed to shed light on the effectiveness of ginger in human subjects with diabetes.

## 4. Conclusions

Diabetes mellitus (DM) is a serious chronic metabolic disease that is characterized by hyperglycemia resulting from defects in insulin secretion, insulin action, or both [113,114]. Insulin is key to maintaining an average level of blood glucose. Insulin production is either absent or decreased in diabetes patients, leading to hyperglycemia [115,116]. Uncontrolled DM can lead to serious chronic complications such as blindness, heart failure, eye problems, stroke, nerve damage, dental disease, and kidney failure [117,118,119]. The treatment strategies for DM have improved over the last few decades. Despite the improvement of patients with DM over the few decades, anti-diabetic drugs have serious side effects such as hypoglycemic coma, and liver and kidney disorders [10]. This review paper highlights the medicinal properties of *Allium sativum*, *Momordica charantia*, *Hibiscus sabdariffa* L., *Zingiber officinale*, and Vitamins (C, D, and E) which possess antidiabetic activities. The antidiabetic activities of these medicinal plants and vitamins are attributed to the presence of coumarins, flavonoids, polyphenols, terpenoids, and other bioactive compounds which exert their effects through the reduction in blood glucose levels. The administration of medicinal plants and vitamins in proper dosages has been demonstrated to lower fasting blood sugar levels in people with diabetes, improve blood circulation and promote wound healing in people with diabetes, and ameliorate complications associated with diabetes [120,121]. Hence, great efforts should be made to increase awareness about the health benefits of medicinal plants and vitamins in the cost-effective prevention and treatment of DM patients; especially those in developing countries who cannot afford the high costs of modern medicine.

## Figures and Tables

**Figure 1 ijms-24-09085-f001:**
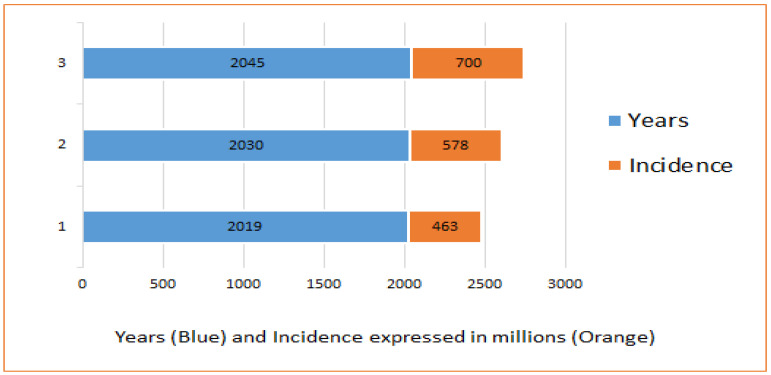
Projection of an increased incidence of diabetes patients worldwide.

## Data Availability

The data that support the present review article are included in the article.

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
