# Peer review of "The Management of Diabetes Mellitus Using Medicinal Plants and Vitamins"

_ijms, 2023, doi:10.3390/ijms24109085_

Round 1

Reviewer 1 Report

This is an interesting and clear review on alternative DM management. The subject was worth addressing. The information is new and useful. Presentation of data is clear, balanced and up to date.

Author Response

Dear Reviewer,

We would like to thank the reviewer for the thoughtful feedback and helpful comments.  To address the issues that were raised, we have streamlined and focused the manuscript considerably.  We have also gathered and compiled additional data so the changes that have been made are substantial and are intended to address all of the issues raised by the reviewer.  We trust that you will find that this is a significant improvement to the review. 

Specific comments

Reviewer: This is an interesting and clear review of alternative DM management. The subject was worth addressing. The information is new and useful. The presentation of data is clear, balanced, and up-to-date.

Response to Reviewer Comment: The reviewer’s input is extremely helpful, and we appreciate the reviewer's positive feedback.

Reviewer 2 Report

1. Scientific names should always be in italics (even in references)

2. Author name for species should be given at the beginning, but not later in the text. Allium sativum L., Momordica charantia L., Hibiscus sabdariffa L., Zingiber officinale Roscoe. Later in the text give only species name not author name.

3. Zingiber officinale  spelling at some places is given wrongly. I corrected the same. Please correct the same in Figure 2 also.

4. Other corrections are marked in the text. Please correct the same.

Please revise and resubmit 

Author Response

Dear Reviewer,

We would like to thank you and the reviewer for the thoughtful feedback and helpful comments.  To address the issues that were raised, we have streamlined and focused the manuscript considerably.  We have also gathered and compiled additional data so the changes that have been made are substantial and are intended to address all of the issues raised by the reviewer.  We trust that you will find that this is a significant improvement to the review. 

Specific comments

Reviewer: Scientific names should always be in italics (even in references)

Response to Reviewer Comment: Thank you for the helpful remarks.  As suggested, we revised the manuscript and make sure that all the scientific names are in italics.

Reviewer: The author's name for species should be given at the beginning, but not later in the text. Allium sativum, Momordica charantia L., Hibiscus sabdariffa L., Zingiber officinale Roscoe. Later in the text give only the species name, not the author's name.

Response to Reviewer Comment: The reviewer’s input is extremely helpful, and we appreciate this feedback. As suggested, we carefully revised the manuscript and have the scientific names of species at the beginning of the text.

Reviewer: Zingiber officinale spelling in some places is given wrongly. I corrected the same. Please correct the same in Figure 2 also.

Response to Reviewer Comment: We thank the reviewer for the helpful comments. To make a better sense, we spelled Zingiber officinale correctly throughout the text.

Reviewer: Other corrections are marked in the text. Please correct the same.

Response to Reviewer Comment: The reviewer’s input is extremely helpful, and we appreciate the reviewer's positive feedback. As suggested, we carefully revised the manuscript and corrected all spelling and grammatical errors.

Reviewer 3 Report

Medicinal plants have been used in medicine since the dawn of time. A medicinal plant is any plant which, in one or more of its organs, contains substances that can be used for therapeutic purposes or which are precursors for the synthesis of useful drugs. This description makes it possible to distinguish between medicinal plants whose therapeutic properties and constituents have been established scientifically, and plants that are regarded as medicinal but which have not yet been subjected to a thorough scientific study. According to data published by the World Health Organization in 2022, approximately 422 million people worldwide have diabetes, with the majority living in low- and middle-income countries, and diabetes is directly responsible for 1.5 million deaths each year. Diabetes has been steadily increasing in both the number of cases and the prevalence over the last few decades. In this sense, the proposed article's theme is timely. In this sense, the proposed article's theme is timely. However, it would be appropriate for the authors to emphasize the choice of plants, emphasizing the importance of these over other medicinal plants. When comparing this article to an identical one that was previously published and was more complete, the few species covered should be given more prominence and scientific relevance. (Bindu Jacob, Narendhirakannan R T. Role of medicinal plants in the management of diabetes mellitus: a review. 3 Biotech. 2019 Jan;9(1):4. doi: 10.1007/s13205-018-1528-0. Epub 2018 Dec 12. PMID: 30555770; PMCID: PMC6291410).

On the other hand, the authors do not explain why they combine medicinal plants with vitamins. Is there any connection between the botanical species and the vitamins? A more detailed introduction is suggested.

The abstract mentions inclusion criteria, results, and discussion, but nothing is referred to as results aside from a figure and a table. The discussion is also missing from this article. 

To summarize, some changes and additional information are suggested to improve this scientific article.

 The manuscript is well written and free of major errors. All bibliography cited in this article adheres to the authors' selection criteria (between 2010-2022).

Author Response

Dear Reviewer,

We would like to thank the reviewer for the thoughtful feedback and helpful comments.  To address the issues that were raised, we have streamlined and focused the manuscript considerably.  We have also gathered and compiled additional data so the changes that have been made are substantial and are intended to address all of the issues raised by the reviewer.  We trust that you will find that this is a significant improvement to the review. 

Specific comments

Reviewer: Medicinal plants have been used in medicine since the dawn of time. A medicinal plant is any plant that, in one or more of its organs, contains substances that can be used for therapeutic purposes or which are precursors for the synthesis of useful drugs. This description makes it possible to distinguish between medicinal plants whose therapeutic properties and constituents have been established scientifically and plants that are regarded as medicinal but which have not yet been subjected to a thorough scientific study. According to data published by the World Health Organization in 2022, approximately 422 million people worldwide have diabetes, with the majority living in low- and middle-income countries and diabetes are directly responsible for 1.5 million deaths each year. Diabetes has been steadily increasing in both the number of cases and the prevalence over the last few decades. In this sense, the proposed article's theme is timely. In this sense, the proposed article's theme is timely. However, it would be appropriate for the authors to emphasize the choice of plants, emphasizing the importance of these over other medicinal plants. When comparing this article to an identical one that was previously published and was more complete, the few species covered should be given more prominence and scientific relevance. (Bindu Jacob, Narendhirakannan R T. Role of medicinal plants in the management of diabetes mellitus: a review. 3 Biotech. 2019 Jan;9(1):4. doi: 10.1007/s13205-018-1528-0. Epub 2018 Dec 12. PMID: 30555770; PMCID: PMC6291410).

Response to Reviewer Comment: We added the scientific report of Bindu Jacob and Narendhirakannan (See Reference 25) in the introduction section to ensure that nothing that has been said is unsupported.  So your inputs have been extremely helpful and we appreciate this feedback. 

Reviewer: On the other hand, the authors do not explain why they combine medicinal plants with vitamins. Is there any connection between the botanical species and the vitamins? A more detailed introduction is suggested.

Response to Reviewer Comment: The reviewer’s input is extremely helpful, and we appreciate this feedback. We agree with the reviewer that we did not explain why they combine medicinal plants with vitamins. The combination of medicinal plants with vitamins was not our focus in this study. Our interest was to highlight individual medicinal plant that shows an anti-diabetic property. It was also focused on highlighting individual vitamins or in combination against diabetes mellitus.

Reviewer: The abstract mentions inclusion criteria, results, and discussion, but nothing is referred to as results aside from a figure and a table.

Response to Reviewer Comment: Thank you for this feedback.  We are not sure that we agree with the reviewer. Our abstract contains a concise summary of our review paper including the background information, rationale, research objective, methodology, result, and concluding insight.  

Reviewer: The discussion is also missing from this article. 

Response to Reviewer Comment: Thank you for this feedback.  We are not sure that we agree with the reviewer.  Our study included a “RESULT and DISCUSSION” section. Please see page 3 of our manuscript for details on the DISCUSSION.

Reviewer: To summarize, some changes and additional information are suggested to improve this scientific article.

Response to Reviewer Comment: Thank you for this feedback.  We have addressed many of the points that you have raised by giving more prominence and scientific relevance to all the species covered in the manuscript.

Comments on the Quality of the English Language

Reviewer: The manuscript is well-written and free of major errors. All bibliography cited in this article adheres to the authors' selection criteria (between 2010-2022).

Response to Reviewer Comment: Thank you for this feedback.

Reviewer 4 Report

The work submitted for review is very important. Diabetes is a non-infectious chronic disease that has become an epidemic of the 21st century. The review proposed for review is of application nature. It can be used, for example, in the formulation of nutritional strategies as a non-pharmacological model of treatment. Currently, when creating recipes for functional food products, the knowledge of enriching these products with biologically active substances facilitates the implementation of modifications to the diet and lifestyle, the change of which is of the greatest importance in the secondary prevention of type 2 diabetes.

The authors did not avoid a few stylistic mistakes. However, in the opinion of the reviewer, the work is cognitive and applied, at the same time it helps to understand the chemistry of active compounds present in herbs.

I would suggest the authors to supplement the review with publications from 2018-2022. This will certainly enrich the work with the latest scientific reports. The development of research in the last decade related to the use of nutraceuticals in the diet therapy of type 2 diabetes is very intensive and the reviewed publication may facilitate the use of this type of products in planning and modeling a non-pharmacological model of treatment of various types of diabetes.

no

Author Response

Dear Reviewer,

We would like to thank the reviewer for the thoughtful feedback and helpful comments.  To address the issues that were raised, we have streamlined and focused the manuscript considerably.  We have also gathered and compiled additional data so the changes that have been made are substantial and are intended to address all of the issues raised by the reviewer.  We trust that you will find that this is a significant improvement to the review. 

Specific comments

Reviewer #4: The work submitted for review is very important. Diabetes is a non-infectious chronic disease that has become an epidemic in the 21st century. The review proposed for review is of an applied nature. It can be used, for example, in the formulation of nutritional strategies as a non-pharmacological model of treatment. Currently, when creating recipes for functional food products, the knowledge of enriching these products with biologically active substances facilitates the implementation of modifications to the diet and lifestyle, the change of which is of the greatest importance in the secondary prevention of type 2 diabetes.

The authors did not avoid a few stylistic mistakes. However, in the opinion of the reviewer, the work is cognitive and applied, at the same time it helps to understand the chemistry of active compounds present in herbs.

I would suggest the authors supplement the review with publications from 2018-2022. This will certainly enrich the work with the latest scientific reports. The development of research in the last decade related to the use of nutraceuticals in the diet therapy of type 2 diabetes is very intensive and the reviewed publication may facilitate the use of this type of products in planning and modeling a non-pharmacological model of treatment of various types of diabetes.

Response to Reviewer Comment: This is a good point and we thank the reviewer for the great remark. We added a few updated citations and take into consideration the reviewer for future publications. (New citation is highlighted in yellow in the text). We trust that you found these updated citations will be self-evident in this revision.

Round 2

Reviewer 3 Report

Taking into consideration the authors' changes, it appears that the scientific article is ready for publication.

Author Response

The reviewer’s input is extremely helpful, and we appreciate this positive feedback.
